# OpenReview forum: "Towards Visually Plausible Explanations"
_ML_Reproducibility_Challenge/2020 — Reject_

### Official Review · AnonReviewer2 · 2021-03-01
**Good reproducibility report**

**Rating:** 7
**Confidence:** 4

**Review:**

***Reproducibility Summary***
The authors provide a complete and useful reproducibility summary.

***Scope of reproducibility***
The authors clearly state the experiments they are trying to reproduce from the original paper and their setup.

***Code***
The authors reimplemented some code and referred to some parts of the original code, which is not complete and is missing parts to reproduce certain experiments.

***Communication with the original authors***
The authors of the report clearly indicate that they have had limited communication with the original authors of the paper through GitHub issues. Since there were some issues in reproducing some of the results, more communication is encouraged to further investigate the issues.

***Hyperparameter search***
The authors coducted reasonable hyperparameter searches considering their computational resources and the amount of details provided in the original paper.

***Ablation study***
The authors have conducted additional ablation studies on the idea proposed in the original paper as an extension of that work.

***Discussion on results***
The discussions of the results is through, highlighting what was reproduced, what was not and possible causes.

***Results beyond the paper***
The authors have included ablation studies as additional results beyond the original paper.

***Overall organization and clarity***
The report is well organized and it is quite clear.

Overall this is a good report, with reasonable experiments that highlight issues in reproducing some of the experiments in the original paper. More communication with the authors is encouraged to try to understand better whether the original results are not correct or there are missing details in this reproduction.

**Familiar With The Original Paper:**

I have read the original paper

**Reproducibility Summary:**

Report has summary

---

### Official Review · AnonReviewer3 · 2021-03-01
**ML Reproducibility challenge - Towards Visually Plausible Explanations**

**Rating:** 4
**Confidence:** 1

**Review:**

The authors reproduce the paper "Towards visually explaining variational autoencoders" [Liu et al. 2020]. The authors provide a brief summary of the paper and try to reproduce all the experiments mentioned in the paper, however for many of the critical experiments the results reported are different from the original paper. For instance, for the MVTec-AD dataset, the authors report numbers in table 2 and for some of the instances, they have double-digit differences in their AUROC measurements.
In addition, some of the provided figures are not very informative. For example in Fig.2 and Fig.3 qualitative results of the reproduced model as well as the original model are provided however since the frames are not exactly paired it is sometimes hard to do a fair comparison. Furthermore, for some frames that look similar, the attention amps look vastly different.
Furthermore, the language of the report can also be improved.
Also an extension of the original paper, authors investigate applying this method  to stacked RBMs but as they claim they could not obtain any meaningful and explainable results.

**Familiar With The Original Paper:**

I have read the original paper

**Reproducibility Summary:**

Report has summary

---

### Official Review · AnonReviewer1 · 2021-03-02
**Reproducibility report for paper - Towards visually explaining variational autoencoders**

**Rating:** 5
**Confidence:** 3

**Review:**

In this report, the authors try to reimplement, reproduce and extend results from the paper Liu et al. (2020) - Towards visually explaining variational autoencoders. The original paper takes a step towards visually explaining generative models by using visual attention maps conditioned on the latent space of a variational autoencoder (VAE).

The authors of the reproducibility report some parts of the original authors’ code and did slight hyper parameter tuning. Most of the results are claimed to be reproduced for parts of the original paper for which the code was available, while for some other parts, reproducing results was hard. The authors of this report also did not have an extensive communication with the original authors, except minor ones on Github by opening an issue.

The report not only tries to reproduce the results from the original paper, but also tried an extension of the work by using Restricted Boltzmann Machines. The authors of the report provide a clear description of the datasets, and include a couple of ablation studies which are useful.

Some of the parts which seemed missing or confusing were:
- Include appropriate citations - for example, for Restricted Boltzmann Machines
- Line 72 - Figure 9 and 8 in the appendix are referred to. But these don’t exist in the appendix
- Notation in the Equation 1 can be clarified
- Line 138 - Could be made clearer on which parts of the original code are being referred and including more details directly from the paper here would also help.
- Formatting of Figure 1
- Line 188 - …similar to that of the paper (see 2). —> Seems the referred number “2” is Table 2? This can be properly labeled and linked


Overall, the authors seems to reproduce the results and make an educated hyper parameter selection. Ablation studies and a couple of insufficiencies from the original paper are also highlighted. If the manuscript can be proof-read and typos and missing figures can be fixed, it can be taken into consideration in the final score.

**Familiar With The Original Paper:**

I have not read the original paper

**Reproducibility Summary:**

Report has summary

---

### Decision · Program_Chairs · 2021-03-31

**Decision:**

Reject

**Comment:**

Overall reviews and/or the paper content not good enough for the AC to recommend to the journal.